# Buckwheat Flour (*Fagopyrum esculentum* Moench)—A Contemporary View on the Problems of Its Production for Human Nutrition

**DOI:** 10.3390/foods12163055

**Published:** 2023-08-15

**Authors:** Pavel Skřivan, Diana Chrpová, Blanka Klitschová, Ivan Švec, Marcela Sluková

**Affiliations:** 1Department of Carbohydrates and Cereals, University of Chemistry and Technology Prague, Technická 5, 166 28 Prague, Czech Republic; pavel.skrivan@vscht.cz (P.S.); blanka.klitschova@vscht.cz (B.K.); ivan.svec@vscht.cz (I.Š.); 2Department of Microbiology, Nutrition and Dietetics, Czech University of Live Sciences Prague, Kamýcká 129, 165 00 Prague, Czech Republic; dianach@centrum.cz

**Keywords:** buckwheat, buckwheat flour, mill processing, quality parameters, nutritional benefits

## Abstract

Buckwheat is returning to the countries of Central Europe; there are several reasons for this: firstly, due to its interesting chemical composition (proteins, fibre, and phenolic compounds), which is reflected in its nutritional value and potential health benefits. Secondly, because buckwheat, and buckwheat flour especially, are suitable raw materials for the production of gluten-free foods. Buckwheat flours are classified similarly to wheat flours, but the different anatomy of wheat grains and buckwheat seeds makes this classification partly misleading. While wheat flours are largely produced by one standard process, the production process for buckwheat flours is more varied. For wheat and wheat flours, the basic quality parameters and their required ranges for different types of primary and secondary processing are clearly defined. This is not the case for buckwheat and buckwheat flours, and the definition of the parameters and their ranges that characterize its technological quality remain unclear. The standardization of quality parameters and production processes is likely to be necessary for the potential expansion of the use of buckwheat for food production and, in particular, for bakery products.

## 1. Introduction

Buckwheat (Fagopyrum) together with amaranth (Amaranthus) and quinoa (Chenopodium) belong to the so-called pseudocereals. For buckwheat, the term pseudocereal is well-established and widely used, whereas amaranth and quinoa are sometimes also referred to as Andean crops. Pseudocereal grains can be processed like cereal grains into flours and then into secondary bakery products, but, unlike cereals, they do not belong to the class of grasses [1]. Interest in pseudocereals has grown, firstly because they contain proteins with better amino acid scores than cereals; a significant proportion of resistant starch [2,3], thus having a lower glycaemic index; and some other biologically active substances [4]. In uncooked buckwheat, resistant starch represents approximately one third of the total starch content [2]. After cooking, the proportion of resistant starch is about 7–10% in heat-treated buckwheat products [3]. In addition, pseudocereals are gluten-free and, therefore, represent potential raw materials for the production of gluten-free bread and bakery products. All this applies in full to buckwheat.

Buckwheat belongs to the family Polygonaceae. It is an annual, dicotyledonous plant, and its fruit is a triangular, brown-coloured achene. Two species of buckwheat are grown today, namely, sown buckwheat (*F. esculentum* (Moench)) and Tatar buckwheat (*F. tataricum* (L.)). Sown buckwheat is mainly used for human consumption [1].

Buckwheat is a plant with a 0.5–1.2 m tall, knobbly, reddish stem that branches considerably in the upper third (Figure 1) [5]. The lower leaves are long-petioled and heart-shaped, whereas the upper leaves are sessile. The root of buckwheat is spherical and usually only penetrates the soil to a shallow depth. Root branching depends on the fertility of the soil, but overall, the root system has a relatively small number of fine roots. Buckwheat has small white or pinkish flowers that are grouped in inflorescences of 7–9. A special feature of buckwheat is its differently shaped flowers, with one type having long anthers and short stigmas and the other type having the opposite. The flowering and ripening of buckwheat are staggered over time, and the same applies to the final stages of ripening, which is also gradual, with the fruit ripening from the lowest branches to the top. This poses a considerable problem for harvesting, or rather the homogeneous maturity of the harvested seeds, which is practically unattainable, unlike in cereals. The length of ripening is mainly influenced by temperature. It is a heterogamous plant whose main pollinator is the honey bee. The fruit of buckwheat is a characteristic triangular achene, brown to purple–black in colour, 4–7 × 3–4 mm in size, and the colour of the peeled achenes changes from light green to brown with age [6].

The structure of the achene resembles a cereal grain [2]. The buckwheat outer layers (husk and pericarp) have a hard fibrillar structure and are usually dark brown or black. Removing the husks by grinding produces buckwheat groats. Below the husk, we find the testa and below that the aleurone layer, which is a simple layer of cells with thin walls that forms the outer layer of the endosperm. The endosperm cells are thin walled and mainly contain starch granules [7].

## 2. A Brief History of Buckwheat in Central Europe

Buckwheat is a crop native to the southwestern region of China. According to historical sources, the first mention of buckwheat cultivation dates back 2500 years. Buckwheat continued to spread to Korea, Japan, and westwards through Russia and Ukraine to Central Europe. Later, it came to the USA as people migrated from Europe and Asia. In Central Europe, buckwheat is one of the youngest crops, and its appearance in the area is associated with the invasions of the Mongol and Turkish armies. In the thirteenth century, its cultivation and processing flourished in Germany, Italy, Austria, and the Czech lands. It was mostly grown in the mountain and foothill areas of Bohemia and Moravia, Austria, and Germany. It was mainly used to make porridge but also bread.

Nowadays, buckwheat is grown mainly in the northern hemisphere, and the main producers are Russia followed by China. The largest importer of buckwheat is Japan, where it is the second most important foodstuff after rice [1].

In the centuries that followed, buckwheat from central Europe gave way to more systematically grown bread cereals, especially wheat and rye, and also to barley grown mainly for malting.

The current return of buckwheat to Central Europe and the growing interest from consumers stems both from the increasing interest in gluten-free foods and from new information on the health benefits that buckwheat may bring [8].

## 3. Chemical Composition and Importance of Buckwheat in Human Nutrition

The composition of buckwheat grains corresponds approximately to that of cereals but differs in many respects. The main component is starch (about 60–70%), followed by protein (about 10–12%), dietary fibre (about 10%), lipids (about 3%), and ash (about 2.5%). Buckwheat contains various minor components of nutritional importance, such as polyphenols, d-chiro-inositol, and some vitamins [4]. All these substances contribute to the various positive health effects attributed to buckwheat. The positive health effects that buckwheat may, according to current knowledge, exert are, in particular, antihypertensive effects and effects on blood sugar (glycaemia) levels, and other properties include antioxidant and anti-inflammatory effects, or it may contribute to some extent to the prevention of cancer [9,10].

The following table (Table 1) compares the chemical compositions of the grains of the selected cereals and buckwheat.

### 3.1. Carbohydrates and Starch

The endosperm of buckwheat contains 70 to 80% starch, which consists of 25% amylose and 75% amylopectin. The starch granules of buckwheat themselves are polygonal in shape and are very often found in clusters. Their size is relatively small, ranging from 2 to 14 µm.

The figure below (Figure 2) shows buckwheat starch imaged by scanning electron microscopy. The polygonal shape of the buckwheat starch granules can be seen in the image, as well as small depressions caused by α-amylase activity [10]. The undamaged starch granules have smooth surfaces and integral shapes, whereas the amylase action damages the surfaces of the granules and causes small indentations that are visible in the picture (especially evident on the right side of Figure 2). The next picture (Figure 3) shows the compact arrangement of starch granules in buckwheat flour particles.

The starch granules of buckwheat start to gelatinize at 65 °C, and the peak viscosity temperature is approximately 90 °C. For different flour samples, the temperature of the onset of gelatinization varies between 57 and 67 °C. Compared with corn and wheat starch, buckwheat starch has a wider range of gelatinization temperatures. This is due to the greater heterogeneity of buckwheat starch and may be related to the milling method and the genetics of the crop. In general, the gelatinization temperature of starch is related to the morphologies of the starch granules, amylose content, amylose and amylopectin structures and crystallinities, and differences between samples [10]. Thus, in terms of properties, buckwheat starch has a higher gelatinization temperature than cereal starches, and its rheological behaviour is more similar to potato starch.

Buckwheat contains a high proportion of resistant starch, up to 30% of the total starch content [2]. When buckwheat is made into flour, the starch granules are prone to be more easily hydrolysed by amylolytic enzymes, as mechanical and thermal damage to the starch during milling makes the starch granules more susceptible to enzymatic hydrolysis. However, buckwheat starch granules are not as easily degraded as wheat starch granules even after conventional milling; this is probably due to the structure of buckwheat starch and also because buckwheat contains some substances that inhibit α-amylase activity [14].

### 3.2. Dietary Fibre

Along with starch, buckwheat is an important source of fibre (cellulose and lignin in the pericarp and non-starch polysaccharides in the endosperm as components of soluble fibre).

The total fibre content of buckwheat seeds is less than 30% if the amount of fibre includes resistant starch. Non-starch polysaccharides and lignin account for 10 to 15% of the total weight of the grain, whereas, in buckwheat groats (after removal of the pericarp), the fibre content without resistant starch is most often in the range 3 to 6%. The amount of fibre in buckwheat flour varies depending on the grain processing method [15,16]. Water extractable dietary fibre accounts for about 50% of it or 1.5–3% of the dry matter of the flour. Water extractable fibre components can exhibit prebiotic effects [4,17,18]. Buckwheat outer layers contain, among other things, significant amounts of phagopyritols (indigestible oligosaccharides), galactosyl derivatives, and *O*-methyl-chiro-inositol, which may assist in the prevention and treatment of type 2 diabetes [2].

In general, foods with higher fibre and resistant starch content have lower glycaemic index values. Buckwheat flour is, therefore, more suitable for diabetic patients than wheat flour [17].

### 3.3. Proteins and Amino Acids

The protein content of buckwheat seeds ranges from 8.5 to 19% and depends on the variety, with the average most commonly reported to be between 10 and 12% [8,19].

Buckwheat protein has a relatively high nutritional value due to its relatively balanced content of individual amino acids, including essential amino acids. While lysine is the limiting amino acid in most cereals, it is relatively abundant in buckwheat [19]. Similarly, compared to cereals, buckwheat contains higher amounts of arginine and aspartic acid and, conversely, lower amounts of proline and glutamic acid [8]. The endosperm storage proteins of buckwheat seeds are composed of 70% globulins, 25% albumins, 4% glutelins, and almost no prolamins [7], which is quite different from cereals, where prolamins and glutelins predominate in the endosperm (in wheat, endosperm storage proteins are composed almost exclusively of these two fractions).

### 3.4. Lipids

The lipids found in buckwheat are mostly concentrated in the germ, which contains, on average, about 6.5% lipids, whereas less than 0.4% is found in the endosperm [8]. The main component of the neutral lipid fraction is triacylglycerols. The most abundant fatty acids are oleic, linoleic, and palmitic acids. These three fatty acids account for up to 90% of the total fatty acid content of buckwheat. Of the total fatty acid content, buckwheat contains 80% unsaturated fatty acids, of which more than 40% are essential linoleic acid [2]. The average levels of unsaturated fatty acids are as follows: oleic acid (580 mg/100 g) and linoleic acid (530 mg/100 g), and α-linolenic acid (80 mg/100 g) is also present in smaller amounts, which is highly beneficial for health, especially in the prevention and treatment of cardiovascular diseases [15,20].

### 3.5. Minerals and Vitamins

Buckwheat contains a variety of mineral substances that are important for human nutrition, most of which are found mainly in the outer layers and germs. The total proportion of these mineral substances in the buckwheat grain is usually around 2.5%, and more than half of them are found in the germ. The main macro-elements present are phosphorus, potassium, magnesium, and, to a lesser extent, calcium, and the microelements are iron, zinc, manganese, and copper [1,2]. The main vitamins present in buckwheat seeds to a greater extent are vitamins belonging to the B vitamin group, namely, vitamin B_1_ (thiamine), B_2_ (riboflavin), and B_3_ (niacin) [18]. Thiamine accumulates most in the aleurone layer, riboflavin in the endosperm, and niacin is most abundant in the envelope layers. In addition, vitamin E and C are also found in buckwheat [1,21].

### 3.6. Phenolic Compounds

There are significant amounts of phenolic compounds in buckwheat. One of the important members of this group is rutin, and buckwheat is an important source of it. The amount of rutin in the whole plant is on average 1.8% of the plant weight and is highly dependent on the growing conditions and variety. The rutin content of buckwheat organs gradually decreases from the leaves to the flowers and stem to the seed. In the achenes themselves, its content is several times lower than in the filaments, and most of it is found in the pericarp. Quercetin and other polyphenols such as orientin, vitexin, isovitexin, and isoorientin are also present [1]. Polyphenols are found in cereals and pseudocereals in both free and bound forms. In buckwheat flour, phenolic compounds are predominantly found in free forms, up to over 90% of them [22].

With the presence of phenolic compounds and a significant proportion of non-starch carbohydrates with prebiotic activities, dark buckwheat flours in particular demonstrate positive health effects. In addition to the aforementioned low glycaemic indexes of buckwheat-based products, their effects on cholesterol levels, and a certain ability to lower blood pressure, their antioxidant effects and abilities to scavenge free radicals are significant. Thus, buckwheat products have an anti-inflammatory effect and, to some extent, may be involved in reducing the probability of colon cancer [10,23].

### 3.7. Antinutrients

Some buckwheat proteins have been found to cause allergies, but little is still known about these properties. In particular, a protein with a molecular weight of 24 kDa. This is converted during cooking into a 30–35 kDa protein, which can also cause an allergic reaction. Other proteins with the potential to cause an allergic reaction include those with the following molecular weights: 69–70, 19, 16, and 9 kDa [1]. Other antinutritional substances that are more abundant in buckwheat achenes themselves include phytic acid, tannin, and protease inhibitors. In addition, phagopyrin is found in buckwheat hulls. This is a phototoxic derivative of hypericin, which, upon reacting with sunlight, causes a disease called phagopyrism. This disease manifests itself in digestive and nervous disorders, eczema, and blisters. Phagopyrin is not present in buckwheat seeds, even in the pericarp, and, therefore, it is more of a threat to livestock that are fed buckwheat sprouts [1].

## 4. Production of Buckwheat Flour

There are at least several basic types of buckwheat flour, and, unlike wheat flour, there is no single definition. Similarly, different milling techniques and mill technologies can be used to produce buckwheat flours.

The strong tough pericarp and the testa are not firmly attached to each other in the mature grain. The pericarp, or fruit sheath, which forms the outer layer of the pericarp, contains a large amount of fibre with a high cellulose and lignin content and is usually dark brown or black in colour [7]. Immediately below the pericarp is the testa, which is loosely attached to the pericarp in mature grain. Below the pericarp is the aleurone layer, and below this is the endosperm, which makes up approximately 70% of the weight of the achenes and in the centre of which is the embryo. The percentage of pericarp is 20–25%, the testa is 1.5–2%, the aleurone layer occupies 4–5%, and the germ, which is located inside the endosperm, unlike in the cereals, accounts for 15–20% [1].

The primary processing of buckwheat generally involves similar technological steps as with cereals. The main stages of buckwheat processing in the mill are the reception of buckwheat seed, storage, cleaning, dehulling, actual grinding, and finishing of the products. The most important quality characteristics of buckwheat groats are colour and taste. As was mentioned above, the colour of the pericarp of freshly harvested seed is light green but gradually changes to reddish brown during storage [11,24].

Buckwheat achenes should contain up to 14% moisture, a maximum of 5% adulterants, and up to 1% impurities. They can be milled either hulled or unhulled, and, before milling, the achenes are sorted on screens according to size (3.2–5 mm holes), as the yield of the product is directly dependent on their uniform size while minimising waste. A major problem posed by buckwheat, unlike cereals, is its gradual ripening. This means that the harvesting process produces a variety of ripeness levels and thus size and composition. The most valuable fraction for sorting is the achenes with a size above 4 mm, and, on the other hand, achenes that are too small with a size below 3 mm are no longer marketable because they are crushed during the hulling process [1].

The freshness of buckwheat grains is one of the leading indicators of quality. The shelf life of hulled buckwheat is significantly lower than that of cereal grains. Long-term storage results in the loss of important sensory substances affecting the typical aroma and taste of buckwheat. To a certain extent, the above-mentioned colour changes also occur, with the light-green fresh grains turning pinkish to brown over time. These qualitative changes, which are mainly due to oxidation processes, can be partially reduced when the seeds are stored at lower temperatures and at relative humidity below 45% [1]. Alternatively, to reduce quality changes, grains can also be stored in a modified atmosphere under a high nitrogen concentration (97%) and controlled access to oxygen (1.5%) and carbon dioxide (1.5%) [25].

The traditional approach to the primary processing of buckwheat is to separate the pericarp and then process the hulled buckwheat grain.

Two different technological processes are used to remove the hard, dark pericarp. The first involves the cold mechanical dehulling of buckwheat hulls, traditionally using grinding stones or rough-surfaced dehulling machines. This method of hulling does not affect the nutritional or taste characteristics. The yield of flour using this method is lower: 45–60%. The second method is hydrothermal dehulling, in which the achenes are steamed and then dried. The steaming takes place in special pressurised chambers or steamers, and the buckwheat achenes are dried quickly after steaming, so that the pericarp breaks and can then be easily separated from the endosperm. This method usually achieves a higher yield of 60–65%, but the moisture and temperature adversely affect the taste and nutritional properties, and also the aforementioned change in the colour of the testa from green to pinkish brown [1,26]. Apart from these two technologies, there are other possible modifications, but they are based on these two basic ones and are a combination of them [27].

In the first stage of primary processing, two fractions are thus usually produced, the husk and the hulled buckwheat grain, on the surface of which the testa and the aleurone layer are preserved to varying degrees. Buckwheat flour is obtained from the hulled grain by disintegration, which can be carried out either in impact mills (crushers), roller mills, or a combination of the two.

The milling process usually results in the following: light or dark flour (depending on the proportion of sub-outer layers retained on the surface of the hulled grain), semolina, and by-products (buckwheat bran). The milling fractions contain varying proportions of endosperm, germ, and outer or sub-outer layers, and the composition of each fraction may vary. The light flour contains mainly endosperm, the semolina consists of hard pieces of endosperm, and the bran contains the sub-outer layers and germ. In terms of milling yield percentages for the different fractions of buckwheat achenes, this is usually 50–65% being flour, the remainder being by-products (buckwheat bran), with a distinction sometimes made between bran (testa and germ) and husks (pericarp) [12,26].

In China and elsewhere in the world, in addition to dry milling, i.e., some of the above methods, wet milling is also used. In the wet milling method, the buckwheat achenes are macerated in water and then the resulting mass is ground, centrifuged, and dried, and, after drying, it is ground again into flour. Evidently, the different milling methods have a significant effect on the physicochemical and functional properties of the flours, and the different methods are adapted to the natures of the products that are subsequently made from the flours [28].

The definition of the different types of buckwheat flours is not uniform. If analogous terminology to wheat flours is used, then it would be correct to only refer to those flours that contain all the anatomical parts of the pericarp, including the pericarp, as whole-grain flours. However, special disintegration procedures must be used to produce such flours in order to ensure that the pericarp, in particular the spikelet at the individual apices of the triangular spikelet, are sufficiently broken down, but, at the same time, to avoid major disintegration and mechanical and thermal damage to the starch granules. This flour has a nutritionally superior composition to light (white) buckwheat flour, precisely because it contains husks that are rich in protein, dietary fibre, and minerals and also contains more significant amounts of rutin [7,29,30].

Due to the technological complexity of the optimal disintegration of the whole achene, in the vast majority of cases, the procedure is as outlined above. The pericarp and, to varying degrees, the testa and aleurone layer are removed from the achene by one of the methods described above. The hulled grain is then ground into a light buckwheat flour consisting mainly of endosperms. Light buckwheat flours are used in bread and pastry production and have the advantage over wheat flours of the absence of gluten and a high proportion of resistant starch.

However, if the husk (pericarp) is carefully removed, the testa can be left virtually intact, and the (unabraded) hulled grain can be milled into dark buckwheat flour, which is also often described as wholegrain flour, which is incorrect when using the analogy with the terminology of cereal flours. Dark buckwheat flours are sensorially more pronounced, which may not be desirable in bread, pastry, or pasta production, but they are nutritionally richer than light buckwheat flours [7,29].

A wide variety of final buckwheat products are produced from hulled buckwheat grains and flours in the secondary processing process. In Japan, where buckwheat is the most important food crop next to rice, traditional dishes made from buckwheat flour include ‘soba noodles’, which are prepared by slicing a thinly spread dough of buckwheat flour and water. Apart from these noodles, buckwheat products such as various biscuits, cakes, tea, and moonshine are common in Japan. Buckwheat noodles are not only found in Japan but also in other Asian countries and in Italy. In some European countries, the consumption of hulled buckwheat grains or groats as a side dish is more prevalent. These countries include, for example, Russia, Ukraine, and Slovenia. In Europe and America, however, buckwheat flour is increasingly being used to make bread and bakery products of all kinds. This is due to the absence of gluten and a greater awareness of buckwheat’s other nutritional benefits, in particular its low glycaemic index. Buckwheat is very popular in France, where buckwheat flour is traditionally used to make savoury buckwheat pancakes (*les galettes*). In France, even beer is known to be made from buckwheat flour, and the first buckwheat beers were brewed in Central Europe, including the Czech Republic. Buckwheat husks and sprouts are traditionally used to make buckwheat tea in Japan and Korea, which has spread to Europe in various forms and mixtures with tea or other herbs [31].

Processes for the production of gluten-free bread have also been developed for the production of buckwheat sourdoughs [32,33,34].

## 5. Problems in Determining the Technological Quality of Buckwheat Flours

With wheat, there is a very sophisticated methodology for checking the technological qualities of grains for primary processing (milling) and flours for secondary processing (baking or pastry). There are clear parameters that describe the characteristics of wheat for different uses and for its classification and grading, generally using baking grades or classes A, B, C, and E (elite) and groups K or D for special baking wheat. For wheat, both the quantity and quality of protein and starch are decisive and, related to starch, also the activity of amylase. On the basis of these quality parameters, wheat is then processed into flours for various purposes, and the methods for checking and defining the technological qualities of the flours are highly sophisticated. These methods are based on the determination of gluten quantity and its properties and the state and degree of starch deterioration. In addition to the analytical parameters, rheological and viscometric methods used to describe and model the behaviours of doughs and flour suspensions are extremely important [11,26,35]. On the basis of these parameters, it is then possible to predict with relatively high probability the behaviour of flours and doughs in secondary processing.

No such parameters have been established for buckwheat, making prediction of the behaviour of buckwheat flours in bread and pastry production very difficult. As the endosperm proteins of buckwheat do not form a similar viscoelastic structure to wheat prolamins and glutelins, and these fractions are only present in buckwheat to a marginal extent, the starch status and associated amylase activity are essential for predicting technological quality.

The starch state is already determined by the native structure of the starch granules and their maturity in the mature grain/achene. It is also influenced by the harvesting and storage conditions, during which partial activation of the amylase and a certain degree of hydrolytic damage to the starch occur. Another important factor is the actual disintegration process and its intensity. The starch granules are subjected to mechanical and thermal impact during milling [11].

The degree of starch damage can be influenced by the intensity of milling, rotor speed, and type of impact mill or cylindrical bench conditions during milling, milling time, temperature increase during milling, and also the technology of raw material preparation before milling [36]. The main feature of mechanically damaged starch is the disturbed microstructure of the starch granules. The starch granules of undamaged starch are usually smooth and have a regular shape characteristic of the crop. Mechanically damaged starch, however, has granules that are distorted, with rougher surfaces or depressions and fissures. A certain degree of starch damage is favourable for increasing water binding and for optimal fermentation processes, in particular for bakery processing [2,36,37,38].

With wheat and wheat flours, the standard methods used to investigate starch condition and amylase activity are mainly the quantitative reduction and amylographic evaluation. Ranges of values are defined that predict the suitability of using wheat or wheat flour for different types of further processing. This is not the case for buckwheat. Both of these methods can be used for buckwheat flour, but different weights of material have to be chosen and the values range over a wider range, without clearly defined technological optima [26]. This is not only because of the different nature of buckwheat starch but also because while the wheat grains in a given batch generally reach a very similar degree of maturity and, therefore, starch granules maturity, this is not the case for buckwheat. Although buckwheat achenes intended for milling are selected on the basis of size (see above), the variance in the degree of maturity of the sorted batch is still relatively broad. The uncertainty in predicting the behaviour of flour during technological processing in the production of bread and pastry from buckwheat flours presents a complication which needs to be addressed.

## 6. Conclusions

Buckwheat is a crop with tradition and great processing potential. In the last few decades, there has been a growing interest in gluten-free cereals and pseudocereals. In addition to being gluten-free, buckwheat has an interesting nutritional benefit, especially due to the low glycaemic index of buckwheat products. While gluten intolerance (especially celiac disease) affects a minority of the population, and there is not yet sufficient evidence to draw conclusions about its spread, type 2 diabetes mellitus is a global epidemic. Buckwheat has specific sensory characteristics that are reflected in flours and secondary products depending on the production technology, the processes of which can be very diverse.

Therefore, in order to make buckwheat more tractable for industrial production, it will be necessary to establish a methodology for predicting its technological properties. The focus is on the characterisation of the starch state and amylase activity by methods applicable in practice and on the definition of parameters that determine the optimal procedures for the use of buckwheat for primary processing, as well as of primary products (flours, groats, and hulled achenes) in secondary production processes.

## Figures and Tables

**Figure 1 foods-12-03055-f001:**
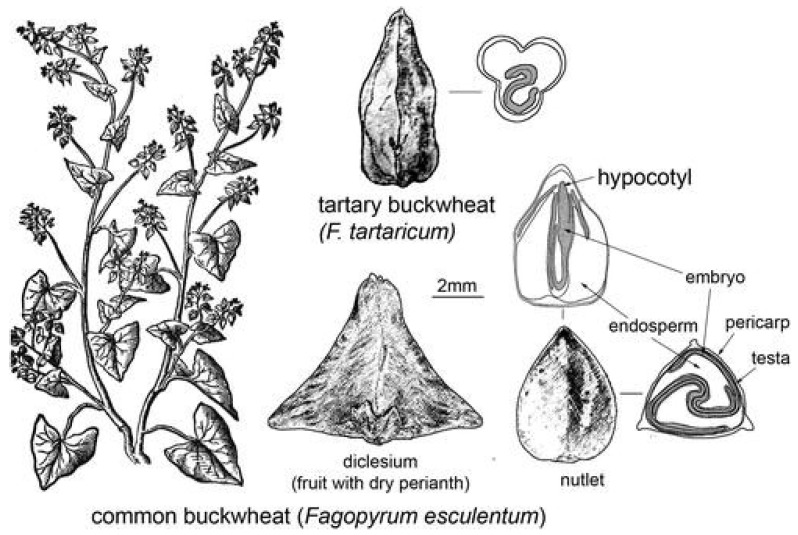
Buckwheat and its parts (plant and achene) [5] (Reprinted with permission from Ref. [5]. Copyright year 2014, copyright Weisskopf and Fuller).

**Figure 2 foods-12-03055-f002:**
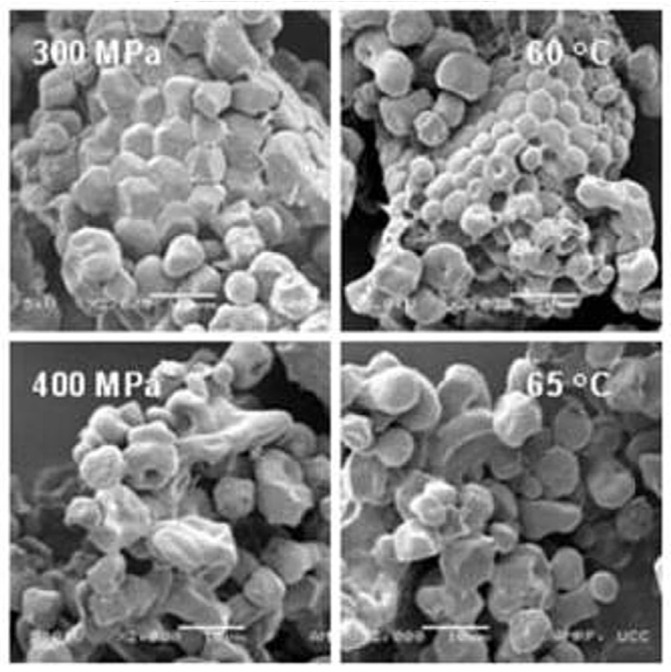
Buckwheat starch observed by scanning electron microscopy [10] (Reprinted with permission from Ref. [10]. Copyright year 2016, copyright Zhu).

**Figure 3 foods-12-03055-f003:**
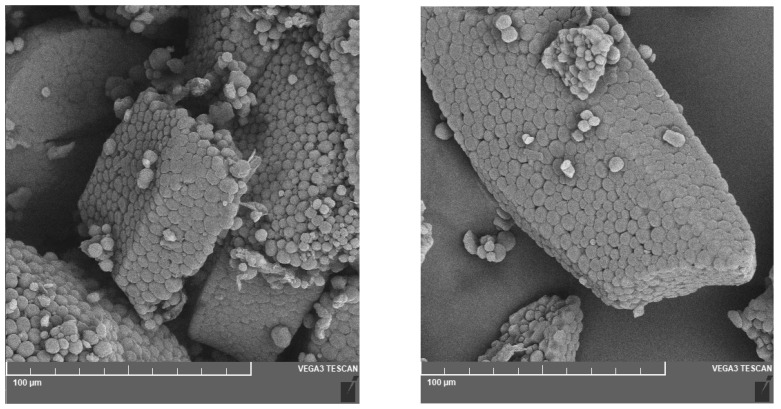
Buckwheat starch—white buckwheat flour observed by scanning electron microscopy [13] (Reprinted with permission from Ref. [13]. Copyright year 2022, copyright Vítová).

**Table 1 foods-12-03055-t001:** Average chemical composition (% *w*/*w* in d. m.) of buckwheat, wheat, rye, and oat grains [11,12] (Reprinted with permission from Refs. [11,12]. Copyright year 2017, copyright Sluková et al.; copyright year 2021, copyright Sinkovič et al., respectively).

Crop/Nutrients	Buckwheat	Wheat	Rye	Oat
Proteins	11.3	13.0	12.5	14.5
Carbohydrates	73.6	68.5	60.0	53.0
Total Dietary Fibre	12.1	9.0	14.0	14.5
Lipids	2.1	3.0	2.5	6.5
Minerals	2.1	2.0	2.0	2.0

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
