# Peer review of "Buckwheat Flour (Fagopyrum esculentum Moench)—A Contemporary View on the Problems of Its Production for Human Nutrition"

_foods, 2023, doi:10.3390/foods12163055_

Round 1

Reviewer 1 Report

The manuscript entitled "Buckwheat flour – a contemporary view of its importance in human nutrition" is a good work, however, it is of preliminary in nature. I recommend some major modifications in the manuscript;

1. The abstract is oversimplified; authors need to incorporate more findings which is of quantitative in nature. 

2. The introduction also needs to be enriched. Authors may include the images of Buckwheat to describe its parts

3. In section 3 of the manuscript, the first two paragraphs seems to be unfit. 

4. The chemical composition should be expanded to its individual chemical constitutents (refer 10.1016/j.foodchem.2020.127653 )

5. The work should be expanded to involve pharmaceutical and nutritional studies (additional tables is mandatory)

6. Altogether the entire manuscript content is limited to one table and two figures. It is insufficient. 

There are several grammatical and spelling errors. it needs to be rectified

Author Response

Dears,

Reviewer 2 Report

The review is of good quality and on a very interesting topic; however, some clarifications are needed to improve the manuscript.

1. Based on the title, I think it would be important for the review to go deeper into the foods that are made with buckwheat and not to remain only in the enumeration.

2. Many Latin American researchers do not share the denomination of pseudocereals for quinoa and amaranth, but rather they are called Andean crops; since the prefix pseudo means false or supposed.

3. L32-L34 A citation is needed to indicate where this information was obtained.

4. Throughout the text different protein contents of buckwheat are mentioned, it would be important to give more consistency to the information.

5. Fig. 1. It would be very useful to indicate with an arrow the depressions produced by the action of amylolytic enzymes and to explain if it is damaged starch grains that are attacked.

6. On this point I have the doubt if it is a problem of use of certain words in the different languages, but I believe that something of polygonal in shape (L103-L104) is a flat figure.

Author Response

Dears,

Reviewer 3 Report

1. The title of this review is Buckwheat flour – a contemporary view of its importance in human nutrition, but the review does not highlight the point of "importance of buckwheat flour in human nutrition". 2. The abstract does not summarize what this review mainly tells about buckwheat flour. 3. The third section is too much about the plant characteristics of buckwheat, but less about the importance of human nutrition, and the title does not match.4. In lines 128-130, it is mentioned that buckwheat starch is not easily degraded because of its structure, what is the specific structure caused by.5. The fourth section is more content, logic is more chaotic, can be divided into several subsections like the third section. 6. In lines 219-221, it is mentioned that there are several types of buckwheat flour, which types are they? And this paragraph mentions the different grinding techniques in the following four paragraphs are not mentioned, these paragraphs should be reordered.7. Is there a contradiction between the two statements: “In addition to being gluten-free, buckwheat has an interesting nutritional benefit, especially due to the low glycaemic index of buckwheat products. While gluten intolerance (especially celiac disease) affects a minority of the population and there is not yet sufficient evidence to draw conclusions about its spread, type 2 diabetes mellitus is a global epidemic”? Since buckwheat does not contain gluten, why mention the gluten intolerance of minority?

Author Response

Dears,

Round 2

Reviewer 1 Report

No more comments